# Energy Poverty and Personal Health in the EU

**DOI:** 10.3390/ijerph191811459

**Published:** 2022-09-12

**Authors:** John M. Polimeni, Mihaela Simionescu, Raluca I. Iorgulescu

**Affiliations:** 1Department of Pharmacy Practice, Albany College of Pharmacy and Health Sciences, Albany, NY 12208, USA; 2Institute for Economic Forecasting-NIER, Romanian Academy, 050711 Bucharest, Romania

**Keywords:** energy poverty, health, energy import dependence, panel ARDL model

## Abstract

The aim of this paper is to assess the impact of energy poverty on health in the EU-27 countries for the period from 2003–2020 using Panel Autoregressive Distributed Lag models and generalized ridge regressions. Arrears on utility bills exerts positive long-run effects on capacity to keep the home adequately warm, current health expenditures, and self-perceived health as bad or very bad, but a negative long-run influence on energy import dependency. In the long-term, the population being unable to keep their home adequately warm positively affects self-perceived health as bad and very bad and negatively influences number of cooling days. Current health expenditure has a long-run influence on self-perceived health as bad and very bad and the number of heating days. Positive short-run impacts were observed for energy import dependency, arrears on utility bills, and number of heating days on current health expenditure and the population unable to keep their home adequately warm. People at risk of poverty or social exclusion in different zones had a significant impact on energy poverty indicators. A separate analysis is made for those EU states with the highest energy import dependency and the implications of the results are discussed.

## 1. Introduction

Access to energy is vital for reducing poverty, economic development, and better living conditions. Energy is necessary for cooking, lighting, and heating and cooling. As such, energy poverty is often thought to be caused by low incomes, energy efficiency, and energy prices [1,2,3] and has a great influence on the lives of millions of people [4]. The literature has often focused on energy prices, lack of energy resources, and low incomes [1,5,6,7,8,9,10,11]. As a result, energy poverty is sometimes defined as the inability to afford the required level of heating in the home [1,10,12,13]. With climate change, extreme heating also requires access to energy for cooling purposes [3,14]. Streimikiene et al. [15] illustrated the importance of considering energy poor households in climate change policies aimed at achieving a carbon free energy transition by 2050 because these households often have behavioral barriers to climate change mitigation. Furthermore, Nawaz [16] showed that climate shocks lead to health poverty and higher per capita health expenditures. Others have defined energy poverty as the inability of a household to obtain the required level of energy services for a home [17,18]. Thus, energy poverty is multidimensional with an objective component, where a household’s net income after energy costs is below the national poverty line and energy costs are above the national average, and a subjective component, where the members of the household state they are not warm enough in the cold season [19,20,21,22]. Typically, the literature has focused on energy prices and affordability [1,11,12,13]. However, the concept of energy poverty has evolved as a lack of energy impacting economic and human development such that an energy-poor society will suffer from issues such as bad health, poverty, and illiteracy [22]. This multidimensional perspective does examine numerous indicators, such as health, poverty, education, and the environment. Recent literature has explored the impact of energy poverty on health, particularly in developing countries [22,23,24,25].

The extent of the problem is large, becoming a priority for institutions such as the World Bank and the United Nations. Energy poverty impacts the health of approximately 34 million people [26,27] and is a major problem throughout the European Union. Yet, there remains a vastly inadequate understanding of energy poverty [22]. This paper considers energy poverty to be much more complex than the previously stated definitions; a bi-directional causal relationship where energy poverty is caused by low income, poor energy efficiency, energy prices, and healthcare costs from poor health status, and energy poverty can create poor health status and low educational status [10,25]. Simply put, if an individual does not have adequate heating, cooling, or refrigeration capability, then they are more susceptible to illness; if that person is sick and incapable of working, then they will not be able to earn income to pay for energy resources. This vicious cycle is bi-directional and, hence, the importance of understanding the energy poverty-health relationship.

Energy poverty was mainstreamed in the EU in 2009 when the third Energy Package was ratified and further developed with the 2018 Clean Energy for all Europeans policy [27]. Many of the Central and Eastern European nations have a large stock of inefficient buildings and lower than average household incomes [27] that can be attributed to their centrally planned economies of the past. The transitional stage, started in 1990, has brought new challenges such as the liberalization of the electricity and gas markets, which has increased energy prices considerably, putting cost pressures on energy consumers [20,28]. Furthermore, many social assistance programs were defunded or eliminated, such as consumer subsidies, price controls, and state support of housing, transportation, and services [27,29,30]. Populations, especially children, the unemployed, the elderly, and low-income individuals and families, are especially vulnerable as a result.

This paper extends the previous literature by exploring the bi-directional causal relationship of energy poverty, poor health, and affordable housing and energy prices. The results indicate that this relationship does, in fact, exist and provides insight for public policy to be developed that will effectively combat energy poverty in the EU. The novelty of the paper is also brought by the methodological approach. In addition to panel data models for all the EU-27 countries to assess the impact of personal health on energy poverty, a cross-country analysis is conducted for the countries with the highest energy import dependency. Consequently, besides overall recommendations for the entire EU, a particular profile is built for some of the countries with vulnerabilities in terms of energy import dependency that need more specific policies.

The rest of the paper is as follows. Section 2 provides a literature review. Section 3 presents the materials and methods used, followed by Section 4, which outlines the theoretical framework employed. Section 5 presents the results. Lastly, Section 6 and Section 7 end the paper with a discussion of the results and conclusion.

## 2. Literature Review

Energy poverty was first considered as fuel poverty, examining how livelihoods were impacted as a result [22,31]. Leach [32] showed energy poverty causes low-income households to spend a higher percentage of their household income on energy than middle-income households, and Boardman [33] quantified the percentage for energy poverty as 10% or more. Hills [11] and Moore [34] both built upon these findings to create new metrics of energy poverty. However, each of these measures apply more for advanced economies where affordability is the main issue rather than developing economies where affordability and accessibility are the problem. Therefore, energy poverty is a much more complex issue.

Energy poverty is multidimensional. Boardman [33] states that the interaction of the socioeconomic status of the household, the energy efficiency of the house, and energy prices are the multidimensional factors impacting energy poverty. Nussbaumer and Bazilian [20] developed a multidimensional energy poverty index, building upon the IEA energy development index, to capture a household’s inability to access energy. Recalde [35] hypothesized that the multidimensional factors are deeper and should include structural determinants such as energy, housing, and labor policies, as well as economic and market policies. As a result, energy poverty can have impacts beyond just affordability, accessibility, and usage for heating, lighting, and other necessities.

Energy poverty is not an insular issue, and it is often related to employment status, and food, housing, and utilities costs [18,36,37]. As a result, energy poverty can have significant health implications [3,38,39,40,41]. In fact, there is extensive evidence of the effects of energy poverty on human health [3,42,43]. Gonzalez-Equino [4] found that energy poverty and health are linked because access to clean energy is necessary to maintain or improve the health of individuals [22]. The direct health effects of energy poverty are increased mortality and morbidity rates through respiratory, cardiac, and cardiovascular diseases [3,44,45], impaired mental well-being, and the intensification of existing health problems. Cold temperatures in a household suppress the immune system, increasing the likelihood of infections and minor illnesses. Indirect health impacts include the impairment of childhood learning, emotional well-being, and resilience. Furthermore, dietary choices and opportunities are reduced, and accidents and injuries at home are more likely. Children, women, and the elderly are most susceptible, as are low-income households. Building on this research, Xiao et al. [46] showed that energy poverty has a negative impact on individual development and learning behavior, and that learning behavior and health conditions are correlated with energy poverty.

Energy poverty is a major problem in Europe. The European Commission has made solving energy poverty a key pillar in their transition to green energy [41]. The situation is only likely to worsen due to a transition to green energy, geopolitical energy dynamics, and reliance on Russia for natural gas supplies. There are considerable gaps in the literature about the impact of energy poverty on health at the European level. Only recently has the literature on energy poverty in Europe become more prevalent, focusing on understanding the socio-demographic factors causing energy poverty, emphasizing vulnerabilities related to income, gender and age [8,41,47,48,49], whereas other studies have explored the roles of energy prices and lack of energy efficiency [41,50,51,52]. There are geographic differences as well; energy poverty is largely more problematic in the Mediterranean and Eastern EU member states [17]. Eastern and Southern European countries are found to be more vulnerable because of higher levels of income poverty, inefficient housing quality, poor infrastructure, and governmental instability [31,41,53]. Energy poverty in the north is typically due to a combination of high energy prices, low incomes, and low energy efficiency. There have been few comparative studies to understand the relationship between energy poverty, health, and well-being in Europe [3,53,54,55]. A few studies have shown that energy poverty negatively impacts health in Southern European countries, focusing on a single or few health outcomes [3,17,35,56,57,58]. Bouzarovski and Tirado Herrero [59] found significant regional differences in energy poverty within Poland, Hungary, and the Czech Republic.

Thomson and Snell [3] did a comparative study of thirty-two European countries to illustrate the negative effects of energy poverty on respiratory and circulatory systems, as well as poor emotional well-being. Karpinska and Smiech [60] did a comparative analysis of energy poverty transitions (moving from a state of energy poverty to energy secure, and vice versa) and persistence of consumers in seventeen European countries. They found the probability of being in an energy poverty state is greater than 51%, for households in Bulgaria, Greece, Lithuania, and Romania close to the energy poverty trap. Oliveras and Peralta [17] found uneven patterns of energy poverty in Southern European countries in gender and social class using more healthcare services and medications.

As a result of the policies set forth by the EU, Romania has as one of the five main goals of the Romanian Energy Strategy 2016–2030 [19] to reduce energy poverty and protect vulnerable people. A major obstacle in achieving these goals is that 90% of the residential floor area was built before 1989 [27,61] when there was no incentive to build with good insulation or energy efficiency measures because housing was highly subsidized and energy costs were low [27,62]. Although efforts have been made recently to improve the energy efficiency of apartment blocks within urban areas in Romania, many still are energy inefficient, as are houses in more rural areas. The energy inefficient dwellings cause households to reduce their energy consumption to save money, such as reducing their room temperature for heating or not using cooling for warm days [27,63].

This paper builds on these findings with the unique approach of exploring bi-directional causality. The next section describes the data and research methodology used which outlines the approach to show a bi-directional causal relationship between energy poverty, poor health, and affordable housing and energy prices.

## 3. Materials and Methods

A panel data approach is used to gain insight into energy poverty in the EU. A cross-country analysis is conducted to study insights for countries with high energy import dependence. Panel data models are built for the EU-27 countries (2003–2020), whereas a separate cross-country analysis is conducted for Italy, Malta, Cyprus, Luxembourg, and Ireland because they have the highest energy import dependency.

Panel Autoregressive Distributed Lag models (panel ARDL) are estimated under the hypotheses of heterogeneity, cross-dependence, and non-integration. The time series approach is developed on low volume samples, so Bayesian nonparametric models such as ridge regression are used for Romania, for example. The data used were obtained from Eurostat and the World Bank. The dependent variables are self-perceived health as bad or very bad (total), cooling days, heating days, energy import dependence, and the share of housing costs in disposable household income. The explanatory variables include arrears on utility bills (total) (arrears), current health expenditure (% of GDP), population unable to keep home adequately warm (population), and people at risk of poverty or social exclusion in cities, towns and rural zones.

Self-perceived health as bad or very bad is used because it is a good indicator of overall personal health and a forecaster of morbidity and mortality [17]. The number of cooling days is important for those individuals living in warmer climates [58] as an indicator for health issues such as heatstroke or other heat related ailments. The number of heating days is necessary to include due to the strong relationship between cold temperatures and illness, such as colds, cardiovascular, and respiratory diseases [59]. Energy import dependence is employed to understand how vulnerable to supply disruptions EU countries are and how any interruptions could worsen energy poverty. The share of housing costs in disposable income is an important factor in whether a household is in energy poverty [8,60,64]. The indicator people at risk of poverty or social exclusion have been shown to be a determinant in energy poverty [65,66,67,68,69,70,71]. The population unable to keep their home adequately warm is an important indicator of energy poverty [72,73,74,75]. Current health expenditure as a percentage GDP [76] is included in the analysis to indicate how a country’s healthcare resources are related to energy poverty. People or households that are in arrears on their utility bills are often affected by energy poverty [47,65,77,78,79,80]. These variables are used in the models below.

## 4. Theoretical Framework

The basic panel ARDL model is based on self-perceived health as bad or very bad, arrears on utility bills, and population unable to keep home adequately warm as the explanatory variables.

Equation (1) is the representation of the panel ARDL model used to explain the logarithm of self-perceived health at time *t* (dependent variable). The explanatory variables are the logarithm of the variables: self-perceived health; arrears on utility bills; and population unable to keep home adequately warm at time (*t − l*). The natural logarithm is applied to all data series to have interpretations in terms of elasticities.
(1)$$\begin{array}{c} \log\left(self\;perceived\;health_{it}\right)=\alpha_{i}+\sum_{l=1}^{p}\beta_{0}\log(self\;perceived\;health_{it-l})+\\ \sum_{l=0}^{q}\beta_{1}\log(arrears_{it-l})+\sum_{l=0}^{q}\beta_{2}\log(population_{it-l})+e_{it} \end{array}$$
where *i* is index for country and *t* is index for time, $e_{it}$ is the error, $\alpha_{i}$ are the coefficients that vary across countries, and $\beta_{0}$,$\;\beta_{1}$,$\;\beta_{2}$ are parameters.

After parameterization, Equation (1) becomes:
(2)$$\begin{array}{c} \Delta \log(self\;perceived\;health\left.h_{it}\right)=\alpha_{i}+\Phi_{i}\left(\log\right.(self\;perceived\;health\left.\left.h_{it-l}\right)\right)-\\ \theta_{1}\log(arrears\left.{}_{it-l}\right)-\theta_{2}\log(population\left.{}_{it-l}\right)+\\ \sum_{l=1}^{p-1}\lambda_{il}\Delta \log(self\;perceived\;health_{it-l})+\sum_{l=0}^{q-1}\lambda_{il}^{\prime }\Delta \log(arrears\left.{}_{it-l}\right)+\\ \sum_{l=0}^{q-1}\lambda_{il}^{\prime \prime }\Delta \log(population\left.{}_{it-l}\right)+e_{it} \end{array}$$
where Δ is the variational operator (this operator is applied to each variable of the corresponding data series in the first difference; i.e., the value of the indicator at time t minus value of the indicator at time (*t* − 1), $\alpha_{i},{\;\Phi }_{i}$ are coefficients that vary across countries, and $\lambda_{il},\lambda_{il}^{\prime },\;\lambda_{il}^{\prime \prime }$ are coefficients that vary across countries and in time. *λ*, *λ*′, *λ*″ represent short-term parameters associated to the lagged endogenous variables, arrears on utility bills, and population unable to keep home adequately warm respectively. *θ*_1_ and *θ*_2_ are the long-term coefficients for arrears on utility bills and population unable to keep home adequately warm. The speed of adjustment is measured by Φ*_i_*. 

A particular type of panel ARDL model is considered based on the pooled mean group (PMG) estimator that considers heterogeneous short-run equilibrium across countries in the sample and homogenous long-run equilibrium across these countries [81]. In the short-run, there are different responses to external shocks of various countries, but in the long-run, the tendency is one of stabilization. The main benefit of the PMG estimator is given by its ability to diminish endogeneity.

Table 1 provides the descriptive statistics for the variables in the panel data models. The maximum value for total arrears on utility bills was registered by Greece in 2014, whereas Denmark reached the minimum value in 2008, before the world financial crisis. Malta was the country with the highest energy import dependence registered in 2013, whereas the lowest value was observed for Denmark in 2005. Croatia registered the maximum value for self-perceived health as bad or very bad in 2010, whereas the minimum was observed in Ireland in 2007.

Traditional regression models could provide spurious results on the empirical data because of assumptions that, in most cases, are not verified by economic data series. Therefore, an accurate analysis should be based on a regression model describing all possible data models. A solution to this problem is given by the Bayesian nonparametric approach that allows the construction of flexible models as an infinite mix of regression models based on minimal data assumptions.

Unlike the least squares estimator in the linear regression model, the linear ridge regression model constructs estimates by reduction. In this case, the prediction error and the mean square error are usually improved.

Given the theoretical background of a Bayesian linear regression model described by O’Hagan and Forster [82], for a given data set $D_{n}=\left(X,y\right)$ with $X={\left(x_{ip}\right)}_{nxp}$ and $y={\left(y_{1},\ldots ,y_{n}\right)}^{T}$ and a normal-inverse gamma conjugate distribution for prior density $\left(\beta ,\sigma^{2}\right)$, we have:(3)$$\begin{array}{c} f\left(y|X,\beta ,\sigma^{2}\right)=n_{n}\left(y|X,\beta ,\sigma^{2}I_{n}\right)=\pi \left(\beta ,\sigma^{2}\right)=n_{p}\left(\beta |m,\sigma^{2}V\right)ig\left(\sigma^{2}|a,b\right)=\\ nig(\beta ,\sigma^{2}|m,V,a,b) \end{array}$$
where $n_{n}\left(.|\mu ,\Sigma \right)$ is the probability density function (pdf) for a normal multivariate distribution, $n_{p}\left(.|\mu ,\sigma^{2}\right)$ is the pdf for a normal univariate distribution, *ig*(.|*a, b*) is the pdf for an inverse gamma distribution (*a* is the form and *b* is the rate, 1/*b* is the scale), $nig(\beta ,\sigma^{2}|m,V,a,b)$ is the pdf for a NIG (normal inverse gamma) distribution (the product between a gamma inverse distribution and a normal multivariate as in Lindley and Smith [83]), and $f\left(y|X,\beta ,\sigma^{2}\right)$ is the likelihood function.

The posterior distribution is a combined likelihood function based on sample data and prior distribution.

If the prior distribution $\left(\beta ,\sigma^{2}\right)$ is NIG, in a marginal approach, β has an a priori Student distribution of mean m and the covariance matrix $V1\left(\beta \right)=\frac{b}{a-1}V$ and 2a degrees of freedom. $\sigma^{2}$ presents a prior inverse gamma distribution of average $\frac{b}{a-1}$ and variance $\frac{b^{2}}{{\left(a-1\right)}^{2}\left(a-2\right)}$.

According to Karabatsos [84], the ridge regression model (RR model) is a linear regression model of Bayesian type with normal a priori distribution $n_{p}(\beta |0,\sigma^{2}\lambda^{-1}I_{p})$ for β, conditioned by $\sigma^{2}$. If $\left(\beta ,\sigma^{2}\right)$ has a normal-inverse gamma distribution $nig(\beta ,\sigma^{2}|0,\lambda^{-1}I_{p},a,b)$, all inference procedures corresponding to the traditional Bayesian linear model are also applied for the ridge regression.

Ridge regression allows for a fast estimation of the coefficients by the least squares method, even if the number of explanatory variables (*p*) is very large and when their number is greater than the number of observations. This approach, which involves many parameters, is also fulfilled in nonparametric Bayesian models. In fact, the specification of models with many parameters (to infinity) is done to obtain a robust and flexible statistical inference. Griffin and Brown [85] showed that Bayesian ridge regression is based on a simple anterior structure and has a good predictive performance in many cases. According to Polson and Scott [86], dimensional linear regression models could be described by Levy processes.

Estimates and the posterior probability are calculated for the standardized parameter to fall within a standard deviation of 0. Beta represents the standardized coefficients (posterior means), based on the centered mean and the normalized explanatory variables of mean 0 and variance 1. The coefficients are between −1 and 1 when all explanatory variables are uncorrelated (there is no multicollinearity).

PP1SD represents the posterior probability that a standardized coefficient is at maximum a standard deviation of zero. An explanatory variable significantly influences the dependent variable if PP1SD is less than 0.5.

*b* represents the non-standardized coefficients on the original scale of the variables. Coefficients are obtained based on standardized beta values:(4)$$b=\left[\bar{y}-\frac{\bar{X}\cdot \beta }{SD{\left(X\right)}^{\prime }},\frac{\beta }{SD\left(X\right)}\right]$$
where *b* are the non-standardized coefficients, *β* are the standardized coefficients, *X* are the means of the explanatory variables in the model, and *SD*(*X*) are the standard deviation of the explanatory variables in the model.

The standardized beta values are automatically computed and by applying Formula (4), the non-standardized coefficients on the original scale of the variables are computed.

The first value in *b* is the intercept (constant), and the other values are slopes. The mean *X* is a line vector that contains the means of the explanatory variables. The standard deviation (*X*)′ is a column vector containing the standard deviations of the explanatory variables.

## 5. Results

### 5.1. Panel Data Models

The hypothesis of heterogeneity is checked because the sample includes both developed and developing countries in the EU that present different patterns of economic, energy, and social development. According to the CD Pesaran test for cross-sectional dependence, if all the p-values are lower than 0.05, then the assumption of cross-section independence is rejected at the 5% level of significance (see Table 2). This cross-sectional dependence might be explained by the common regulations applied to all EU countries.

Next, the data are transformed using their logarithmic forms that allow interpretations in terms of elasticities. Under the assumption of heterogeneity, the Im-Pesaran-Shin test is used since the panel is unbalanced. The null hypothesis of this test considers that all the panels present unit roots. As shown in Table 3, the data series are stationary for heating days, cooling days, people at risk of poverty or social exclusion in towns and people at risk of poverty or social exclusion in rural environment.

Since the evidence of stationarity is mixed, the existence of cointegration is checked for the series with the same order of integration. The Westerlund test for cointegration is considered under the null hypothesis that some panels are cointegrated and, respectively, all the panels are cointegrated (see Table 4).

The results indicate no clear evidence for cointegration and panel ARDL models are built in all the cases. Since the Hessian is unstable or asymmetric in some cases, health expenditure is not considered as a control variable in a few of the models. The PMG estimators presented in Table 5 suggest a long-run relationship between the dependent variables and regressors. In the case of a deviation from the long-run, the speed of adjustment to the equilibrium is given by the error correction term in absolute value. The highest rate of correction was registered in the model explaining cooling days, whereas the lowest one in the model explains self-perceived health as bad or very bad.

The results suggest both short-run and long-run impacts on health at the 95% confidence level. The long-run connections will be examined first. If a person or household is arrears on their utility bills, cannot keep their home adequately warm, or has increased current health expenditures then the long-run impact on self-perceived health as bad or very bad worsens. Furthermore, being arrears on utility bills is also related to a household spending more of their disposable income on housing, and a greater number of heating days. However, there is a negative long-run impact of arrears on utility bills on energy import dependency. Other long-run effects were found in the analysis. The results suggest that there is a positive long-run influence of the population unable to keep their home adequately warm on self-perceived health as bad and very bad and a negative long-run impact on number of cooling days. There is also a long-run impact that was found for current health expenditure on self-perceived health as bad and very bad and the number of heating days.

Several short-run influences were also found. There is a positive short-run impact of being arrears on utility bills on energy import dependency and a negative short-run effect on the number of heating days. In addition, in the short-run, those individuals self-identifying as having bad or very bad health perceive their health as improving with an increase in current health expenditures. Positive short-run influences were found for both current health expenditure and the population unable to keep their home adequately warm on the number of heating days.

These long-run and short-run results are expected. An individual or household that cannot pay their utility bills on-time, cannot keep their house to an adequate heating temperature, or has higher health expenditures will have worse health over time [3,18,24]. Moreover, logic dictates that being in arrears on utility bills suggests more disposable income is spent on housing and that there might be more cold days requiring heating. As a result, less disposable income is available to pay bills such as utilities. The bi-directional causal relationships are evident in these results. In the short-run, however, increases in health expenditures can improve health, whereas long-term health expenditures indicate more serious health issues. Furthermore, increased expenses or an inability to pay expenses and keep a home warm will negatively impact health. If there is an inability to keep a home adequately warm, the ability to keep a home sufficiently cool will also decrease, as does the share of housing costs in disposable income.

When additional control variables are checked for robustness (people at risk of poverty or social exclusion in towns, cities and rural zones), the long-run relationships are kept, as shown in Table 6. People at risk of poverty or social exclusion in cities and towns exerted a positive long-run influence on the share of housing costs in disposable household income and cooling days and a negative impact on heating days. People at risk of poverty or social exclusion in rural zones had a direct long-term impact on self-perceived health as bad or very bad, share of housing costs in disposable household income, and cooling days. In the short-term, people at risk of poverty or social exclusion in towns or rural zones had a positive influence on cooling days and a negative impact on self-perceived health as bad or very bad. These results should be expected as poverty and social exclusion are associated with lower incomes [7,87], severely impacting the ability to address health issues, or the ability to pay for appropriate levels of heating and cooling for the household. 

In addition to the panel data analysis, a cross-country study is deemed necessary since the overall conclusions for the entire sample might not be applied for each country. Particular attention should be paid to countries with high energy import dependency for which the impact of previous regressors on dependent variables is assessed.

### 5.2. Cross-Country Analysis

A cluster analysis based on the average values of energy import dependency and a k-means method was done to identify countries with high values of this indicator and states with lower values (Table 7).

A separate individual analysis based on generalized ridge regressions was made for a few countries with the highest energy import dependency, represented in cluster 1: Malta, Cyprus, Italy, Luxembourg, and Ireland.

In Italy (Table 8), being more in arrears on utility bills and an increase in the number of people unable to keep their home adequately warm increases the negative perception on health, whereas being more in arrears on utility bills and a decrease in the number of people unable to keep their home adequately warm contributes to higher energy import dependency. Energy poverty affects approximately three million poor households in Italy [88]. Thus, these results are consistent with expectations as energy poverty, poverty, and inequality are synonymous with one another, and nearly 1.7 million households live in absolute poverty in Italy [89].

The results for Ireland (Table 9) indicate that self-perceived health rated as bad or very bad is explained by the population unable to keep their home adequately warm and people at risk of poverty or social exclusion in towns and a rural environment. Energy import dependency is determined by arrears on utility bills and people at risk of poverty or social exclusion in cities. Furthermore, the share of housing costs in disposable household income is explained by the people at risk of poverty or social exclusion in rural zones.

More people unable to keep their home adequately warm and more people at risk of poverty or social exclusion in rural zones increases the chances that a person will perceive their health as bad or very bad in Ireland. On the other hand, more people at risk of poverty or social exclusion in towns decrease the chances to perceive unsatisfactory health. This result might suggest that poor people in Irish towns feel healthier than those in villages due to better quality of medical services for these people. When arrears on utility bills in Ireland increases, the energy import dependency decreases, whereas more people at risk of poverty or social exclusion in cities increases energy import dependency. More people at risk of poverty or social exclusion in Irish rural zones contributes to the increase in the share of housing costs in disposable household income in Ireland. A generally accepted theory that holds in Ireland is that energy poverty arises out of low income and inefficient homes [90,91], seemingly confirmed by the results of the analysis presented here, as housing in rural areas tends to be less energy efficient and of lower building quality.

In the case of Cyprus (Table 10), the population unable to keep their home adequately warm is positively correlated with self-perceived health as bad or very bad, whereas more people at risk of poverty or social exclusion in towns and rural zones reduces the perceptions of bad or very bad health. More people unable to keep their home adequately warm and more people at risk of poverty or social exclusion in cities increases energy import dependency. More people arrears on bills and more people at risk of poverty or social exclusion in cities increases the share of housing costs in disposable household income. Cyprus has one of the higher percentages of arrears on utility bills in the EU [92] and a large percentage of households with poor energy affordability [93] and buildings that have poor quality of construction and an aging building stock [92].

More arrears on utility bills and more people unable to keep their home adequately warm contributes to the negative perception on health in Luxembourg (Table 11). Energy import dependency increases when arrears on utility bills grow and the population unable to keep their home adequately warm and people at risk of poverty or social exclusion in villages decreases. More people unable to keep their home adequately warm and more people at risk of poverty or social exclusion in towns increases the share of housing costs in disposable household income. These results build upon the findings of Michel [94] and are expected as Luxembourg has one of the most expensive housing markets in the EU.

In Malta (Table 12), more arrears on utility bills and more people at risk of poverty or social exclusion in towns improves the personal perception on health as being bad and very bad. One likely explanation for this is that people use their income to pay for their healthcare needs which can be expensive, thus, making a preference choice for healthcare overpaying for utilities. Nevertheless, energy poverty in Malta has increased a lot over the study time-period [77].

## 6. Discussion

The energy import dependency of the EU has the potential to severely increase energy poverty for its member states. A large percentage of energy imported to the EU comes from Russia. Should there be a dispute, politically or militarily, Russia could easily decrease or stop the flow of natural gas, for example, hurting countries in the Eastern portion of the EU, particularly Germany. The energy supply issue has been shown to be a problem for the EU in the Russia-Ukraine war. The supply of energy will become a bigger issue in the winter months as citizens of the EU will require natural gas to heat their homes.

Additionally, the EU has set policy to reduce their reliance on fossil fuels to decrease carbon emissions. To achieve this goal, the EU will need to substantially increase the use of renewable energy. This, in turn, will cause more volatility in energy markets in the short to medium run as the transition to green energy occurs. Consequently, the transition could negatively impact vulnerable households and individuals and their health. To offset these potential negative impacts from energy import dependency and the transition to green energy, the EU must increase their production of energy and increase budget allocations for public health and payments to offset energy costs. An increase in energy production would decrease their reliance on energy imports while simultaneously keeping the most vulnerable energy consumers from harm from higher energy prices and less supply. Moreover, as energy production is ramped up, public health budgets and allocations for energy assistance must be increased to offset any negative health impacts to residents and to prevent them.

As the EU transitions to green energy, the transition either must be rapid, which is unlikely to be feasible, or done in stages to reduce the harmful side-effects to vulnerable energy consumers. If the transition is otherwise, vulnerable energy consumers would need compensation to offset higher energy costs. These payments would add significantly to the budgets of EU countries, putting additional pressures on economies and deficits. As a result, energy poverty could be a ticking time bomb for the EU.

In response, countries across Europe have initiated public programs to protect the most vulnerable energy consumers, instituting policies to counteract high energy prices, such as tax reductions and cash payments to low-income households. Investments in energy efficiency improvements can reduce household energy consumption, resulting in energy poverty rates [95]. New construction regulations and investments in building renovations improving the minimum energy performance for buildings can help address some of these issues. Al-Tal et al. [96] showed that energy efficiency is vital in reducing energy poverty. Additionally, as suggested by Bukari et al. [97], since energy poverty increases household health expenditures, broadening the scope of health insurance in EU countries will be helpful in addressing negative health outcomes caused by energy poverty. However, the biggest investment, which has lagged for many years in the EU because of a lack of will by EU politicians, will be in developing more energy supply. Nawaz [16] suggested similarly, that to promote a health society, governments will need to increase the provision of energy services. In the short-run, this may require the use of energy resources that are unattractive to EU governments and, perhaps, citizens, but will likely be a necessary evil as the transition to green energy occurs.

## 7. Conclusions

Energy poverty in Europe is widespread, complex, and created by a combination of factors such as high energy prices, low incomes, and poor energy efficiency. The results of this research has expanded the findings of Halkos and Gkampoura [80] that electricity prices, unemployment, and the percentage of people at risk of poverty are drivers for energy poverty, whereas GDP per capita has an inverse relationship with energy poverty. The results have also expanded upon the findings of the literature cited earlier in this paper showing the bi-causal relationship between energy poverty and health. Our findings show that energy poverty leads to consumer vulnerability, impacting the state of a person’s health. These results have many implications as extreme heat and cold temperatures become more commonplace, disproportionately impacting the most vulnerable households, particularly the elderly, women, and the poor. Furthermore, household demand for energy is price-inelastic in the long-run [95]. Therefore, there will be higher levels of energy poverty resulting in higher poverty, reduced food consumption, and increased health issues because households must spend a greater proportion of their income on energy.

Despite the many contributions to the literature this research makes, the paper does not lack inherent limitations. Most importantly, the research presented in this paper was limited by the availability of data. Future research will explore creating an energy poverty index to capture all the meaningful energy poverty indicators into one measurement. This index will ensure that energy poverty can be fully analyzed, whereas individual indicators may not be captured in a significant relationship. This paper has shown the connection between energy poverty, poor health, and affordable housing and energy prices, filling a gap in the literature. The results of this study are important as they illustrate who might be the most vulnerable, enabling decision makers to develop better public policy to combat energy poverty in the EU. Since countries with high energy import dependency are more likely to enhance energy poverty, specific solutions should be adopted for Malta, Cyprus, Italy, Luxembourg, and Ireland based on national policies that might exceed the European framework. From this point of view, our paper is a step-forward in the research of energy poverty in Europe by providing the most effective recommendations for the most vulnerable countries.

## Figures and Tables

**Table 1 ijerph-19-11459-t001:** Descriptive statistics.

Variable	Mean	Standard Deviation	Minimum Value	Maximum Value
Arrears on utility bills (total)	10.44	8.36	4.7	65.4
Heating days	2821.02	1148.24	322.36	6179.75
Cooling days	119.14	182.12	0	812.18
Energy import dependency	57.08	26.57	−50.62	104.14
Share of housing costs in disposable household income (%)	38.41	10.72	11.8	76.5
People at risk of poverty or social exclusion in cities (%)	22.39	5.92	10.4	56.9
People at risk of poverty or social exclusion in towns (%)	22.87	9.03	9.3	62.3
People at risk of poverty or social exclusion in rural environment (%)	22.90	8.66	9	64.4
Population unable to keep home adequately warm	11.39	11.98	0.3	69.5
Self-perceived health as bad or very bad (total)	10.58	4.75	2.5	26.8
Current health expenditure (% of GDP)	8.13	1.72	4.70	11.58

Source: own calculations in Stata 15.

**Table 2 ijerph-19-11459-t002:** The results of CD Pesaran’s test.

Variable	Calculated Statistics *
Arrears on utility bills; total population (%)	21.76
Heating days	46.30
Cooling days	32.76
% Population unable to keep home adequately warm	10.73
Energy import dependency	11.36
Self-perceived health as bad or very bad; total population (%)	4.55
Share of housing costs in disposable household income (%)	18.33
People at risk of poverty or social exclusion in cities (%)	9.21
People at risk of poverty or social exclusion in towns (%)	11.47
People at risk of poverty or social exclusion in rural environment (%)	11.47
Current health expenditure (% of GDP)	20.63

Note: * indicates that all the *p*-values are less than 0.05. Source: own calculations in Stata 15.

**Table 3 ijerph-19-11459-t003:** The results of Im-Pesaran-Shin test for unit roots in panel data.

Variable (in Log)	Calculated Statistics (*p*-Values)
Arrears on utility bills; total population (%)	−0.2600 (0.3974)
Heating days	−7.3388 (<0.05)
Cooling days	−9.2174 (<0.05)
Population unable to keep home adequately warm (%)	−0.0732 (0.4708)
Share of housing costs in disposable household income (%)	0.4908 (0.5567)
People at risk of poverty or social exclusion in cities (%)	−0.2292 (0.4093)
People at risk of poverty or social exclusion in towns (%)	−2.1715 (<0.05)
People at risk of poverty or social exclusion in rural environment (%)	−2.1805 (<0.05)
Self-perceived health as bad or very bad; total population (%)	0.1196 (0.5476)
Energy import dependency	−0.3141 (0.7234)
Current health expenditure (% of GDP)	−1.1948 (0.1161)

Source: own calculations in Stata 15.

**Table 4 ijerph-19-11459-t004:** The Westerlund test for cointegration.

Dependent Variable	Explanatory Variables	Some Panels Are Cointegrated	All the Panels Are Cointegrated
		Computed Statistics (*p*-Value)	Computed Statistics (*p*-Value)
Share of housing costs in disposable household income (%)	Arrears on utility bills Population unable to keep home adequately warm Current health expenditure	2.3843 (0.0086)	−0.3998 (0.3446)
Self-perceived health as bad or very bad; total population (%)		0.8504 (0.1975)	0.8036 (0.2108)
Energy import dependency		−0.6484 (0.2584)	−0.9792 (0.1637)

Source: own calculations in Stata 15.

**Table 5 ijerph-19-11459-t005:** PMG estimators for the EU-27 countries (2003–2020).

Connection	Variables (in Log)	Dependent Variable (*p*-Value)				
		Self-Perceived Health as Bad or Very Bad (Total)	Energy Import Dependency	Share of Housing Costs in Disposable Household Income	Heating Days	Cooling Days
Long-run relationship	Arrears on utility bills	0.149 (0.000)	−0.167 (0.000)	0.362 (0.000)	0.046 (0.000)	−0.028 (0.534)
	Population unable to keep home adequately warm	0.121 (0.000)	0.006 (0.624)	−0.055 (0.042)	0.007 (0.502)	−0.117 (0.056)
	Current health expenditure	1.729 (0.000)	-	-	0.131 (0.017)	-
Error correction term		−0.219 (0.000)	−0.429 (0.000)	−0.493 (0.000)	−0.974 (0.000)	−1.035 (0.000)
Short-run relationship	Arrears on utility bills	0.021 (0.625)	0.066 (0.006)	−0.048 (0.43)	−0.059 (0.090)	−0.380 (0.411)
	Population unable to keep home adequately warm	0.043 (0.150)	0.004 (0.888)	0.065 (0.258)	0.098 (0.009)	0.214 (0.425)
	Current health expenditure	−0.315 (0.023)	-	-	0.235 (0.082)	-
	Constant	−0.469 (0.000)	1.922 (0.000)	1.390 (0.000)	7.324 (0.000)	3.876 (0.000)
Residuals	I(0)	I(0)	I(0)	I(0)	I(0)	I(0)

Source: own calculations in Stata 15.

**Table 6 ijerph-19-11459-t006:** Robustness: PMG estimators for additional variables in the case of EU-27 countries (2003–2020).

Connection	Variables (in Log)	Dependent Variable (*p*-Value)														
		Self-Perceived Health as Bad or Very Bad (Total)			Energy Import Dependency			Share of Housing Costs in Disposable Household Income			Heating Days			Cooling Days		
Long-run relationship	Arrears on utility bills	0.095 (0.000)	−0.022 (0.267)	−0.002 (0.916)	−0.167 (0.000)	−0.193 (0.000)	−0.193 (0.000)	0.070 (0.000)	0.112 (0.000)	0.112 (0.000)	0.19 (0.000)	0.037 (0.004)	0.037 (0.004)	−0.069 (0.231)	−0.124 (0.056)	−0.124 (0.056)
	Population unable to keep home adequately warm	0.043 (0.046)	-	0.020 (0.209)	0.005 (0.711)	−0.00006 (0.997)	−0.0006 (0.997)	−0.195 (0.000)	-	-	−0.149 (0.000)	0.031 (0.006)	0.031 (0.006)	−0.209 (0.010)	−0.167 (0.014)	−0.167 (0.014)
	Current health expenditure	−0.244 (0.02)	-	-	-	-	-	-	-	-	−0.511 (0.000)	-	-	-	-	-
	People at risk of poverty or social exclusion in cities	0.044 (0.507)	-	-	0.009 (0.831)	-	-	0.561 (0.000)	-	-	−0.042 (0.000)	-	-	0.304 (0.090)	-	-
	People at risk of poverty or social exclusion in towns	-	0.331 (0.000)	-	-	0.049 (0.308)	-	-	0.103 (0.000)	-	-	0.002 (0.924)		-	0.284 (0.042)	-
	People at risk of poverty or social exclusion in rural zones	-	-	0.651 (0.000)	-	-	0.047 (0.307)	-	-	0.103 (0.0000)	-	-	0.002 (0.924)	-	-	0.284 (0.042)
Error correction term		−0.356 (0.000)	−0.366 (0.000)	−0.305 (0.000)	−0.378 (0.000)	−0.376 (0.000)	−0.377 (0.000)	−0.467 (0.000)	−0.671 (0.000)	−0.672 (0.000)	−0.508 (0.009)	−0.928 (0.000)	−0.928 (0.000)	−1.020 (0.000)	−1.021 (0.000)	−1.021 (0.000)
Short-run relationship	Arrears on utility bills	0.006 (0.857)	0.099 (0.010)	0.085 (0.027)	0.046 (0.113)	0.089 (0.006)	0.089 (0.006)	−0.012 (0.848)	0.007 (0.882)	0.007 (0.882)	−0.103 (0.005)	−0.007 (0.733)	−0.007 (0.733)	−0.686 (0.080)	−0.546 (0.224)	−0.546 (0.224)
	Population unable to keep home adequately warm	0.076 (0.105)	0.072 (0.002)	0.074 (0.002)	0.006 (0.839)	−0.002 (0.956)	−0.002 (0.956)	0.197 (0.019)	-	-	0.206 (0.003)	-	-	0.219 (0.538)	0.216 (0.376)	0.216 (0.376)
	Current health expenditure	−0.062 (0.682)		-	-	-	-	-	-	-	0.500 (0.013)	-	-	-	-	-
	People at risk of poverty or social exclusion in cities	−0.092 (0.251)	-	-	−0.188 (0.501)	-	-	0.008 (0.929)	-	-	0.070 (0.565)	-	-	0.653 (0.384)	-	-
	People at risk of poverty or social exclusion in towns	-	−0.113 (0.010)	-	-	−0.049 (0.486)	-	-	−0.074 (0.115)	-	-	0.070 (0.304)	0.070 (0.304)	-	0.968 (0.032)	-
	People at risk of poverty or social exclusion in rural zones	-	-	−0.142 (0.001)	-	-	−0.046 (0.485)	-	-	0.008 (0.929)	-	-	−0.073 (0.114)	-	-	0.968 (0.032)
	Constant	−0.800 (0.000)	0.463 (0.000)	0.057 (0.193)	1.688 (0.000)	1.671 (0.000)	1.671 (0.000)	0.959 (0.000)	2.089 (0.000)	2.089 (0.000)	4.583 (0.007)	7.132 (0.000)	7.132 (0.000)	3.079 (0.000)	3.249 (0.000)	3.249 (0.000)
Residuals	I(0)	I(0)	I(0)	I(0)	I(0)	I(0)	I(0)	I(0)	I(0)	I(0)	I(0)	I(0)	I(0)	I(0)	I(0)	I(0)

**Table 7 ijerph-19-11459-t007:** Cluster Membership.

Cluster Distribution		
Country	Cluster	Distance
Greece	1	0.934
Austria		3.420
Lithuania		3.989
Slovakia		5.396
Spain		5.928
Belgium		8.033
Germany		8.261
Portugal		8.714
**Italy**		10.110
Hungary		11.603
**Ireland**		13.857
Latvia		15.205
Finland		19.576
Croatia		19.902
Slovenia		19.942
France		21.122
**Cyprus**		26.185
**Luxembourg**		26.741
**Malta**		29.782
Romania	2	2.107
Poland		3.209
Czechia		3.473
Estonia		8.409
Sweden		8.606
Netherlands		13.094
Bulgaria		15.874
Denmark		33.741

Source: own calculations in SPSS.

**Table 8 ijerph-19-11459-t008:** The results of generalized ridge regressions for Italy.

Variables (in Log)	Dependent Variable Unstandardized Coefficients (PP1SD)				
	Self-Perceived Health as Bad or Very Bad (Total)	Energy Import Dependency	Share of Housing Costs in Disposable Household Income	Cooling Days	Heating Days
Arrears on utility bills	**0.516** **(0.003)**	**0.035** **(0.263)**	−0.0001 (0.668)	−0.035 (0.601)	0.015 (0.608)
Population unable to keep home adequately warm	**0.385** **(0.201)**	**−0.067** **(0.236)**	−0.0001 (0.668)	0.006 (0.667)	−0.023 (0.617)
People at risk of poverty or social exclusion in cities	−0.335 (0.631)	−0.088 (0.572)	0.0001 (0.668)	0.096 (0.644)	0.019 (0.476)
People at risk of poverty or social exclusion in towns	0.428 (0.647)	−0.032 (0.661)	0.0001 (0.668)	0.123 (0.642)	0.014 (0.643)
People at risk of poverty or social exclusion in rural zones	−0.430 (0.649)	−0.034 (0.664)	0.0001 (0.668)	0.125 (0.643)	0.022 (0.644)

Source: own calculations in MATLAB. Bold numbers are significant.

**Table 9 ijerph-19-11459-t009:** The results of generalized ridge regressions for Ireland.

Variables (in Log)	Dependent Variable Unstandardized Coefficients (PP1SD)				
	Self-Perceived Health as Bad or Very Bad (Total)	Energy Import Dependency	Share of Housing Costs in Disposable Household Income	Cooling Days	Heating Days
Arrears on utility bills	−0.022 (0.664)	**−0.193** **(0.348)**	0.162 (0.558)	0.025 (0.555)	0.020 (0.546)
Population unable to keep home adequately warm	**0.145** **(0.401)**	−0.037 (0.644)	0.100 (0.614)	0.011 (0.502)	0.016 (0.568)
People at risk of poverty or social exclusion in cities	−0.112 (0.626)	**0.849** **(0.021)**	0.206 (0.635)	0.023 (0.634)	0.028 (0.624)
People at risk of poverty or social exclusion in towns	**−0.487** **(0.001)**	−0.011 (0.666)	0.205 (0.610)	0.009 (0.657)	0.014 (0.579)
People at risk of poverty or social exclusion in rural zones	**1.522** **(0.002)**	−0.083 (0.656)	**0.253** **(0.452)**	−0.0009 (0.673)	−0.001 (0.669)

Source: own calculations in MATLAB. Bold numbers are significant.

**Table 10 ijerph-19-11459-t010:** The results of generalized ridge regressions for Cyprus.

Variables (in Log)	Dependent Variable Unstandardized Coefficients (PP1SD)				
	Self-Perceived Health as Bad or Very Bad (Total)	Energy Import Dependency	Share of Housing Costs in Disposable Household Income	Cooling Days	Heating Days
Arrears on utility bills	0.015 (0.665)	−0.003 (0.647)	**0.181** **(0.481)**	−0.003 (0.665)	0.001 (0.667)
Population unable to keep home adequately warm	**0.492** **(0.153)**	**0.046** **(0.221)**	0.329 (0.610)	−0.040 (0.554)	0.001 (0.667)
People at risk of poverty or social exclusion in cities	−0.082 (0.662)	**0.053** **(0.346)**	**0.179** **(0.324)**	−0.055 (0.596)	0.001 (0.667)
People at risk of poverty or social exclusion in towns	**−0.499** **(0.006)**	0.01 (0.582)	0.471 (0.552)	0.004 (0.666)	0.001 (0.667)
People at risk of poverty or social exclusion in rural zones	**−0.500** **(0.006)**	0.011 (0.582)	0.489 (0.556)	0.003 (0.667)	0.001 (0.667)

Source: own calculations in MATLAB. Bold numbers are significant.

**Table 11 ijerph-19-11459-t011:** The results of generalized ridge regressions for Luxembourg.

Variables (in Log)	Dependent Variable Unstandardized Coefficients (PP1SD)				
	Self-Perceived Health as Bad or Very Bad (Total)	Energy Import Dependency	Share of Housing Costs in Disposable Household Income	Cooling Days	Heating Days
Arrears on utility bills	**0.080** **(0.096)**	**0.004** **(0.464)**	−0.048 (0.597)	0.194 (0.573)	0 (0.669)
Population unable to keep home adequately warm	**0.077** **(0.265)**	**−0.003** **(0.480)**	**0.119** **(0.201)**	0.116 (0.591)	0 (0.669)
People at risk of poverty or social exclusion in cities	0.047 (0.652)	−0.006 (0.634)	0.179 (0.598)	−0.221 (0.658)	0 (0.669)
People at risk of poverty or social exclusion in towns	0.070 (0.612)	**−0.003** **(0.152)**	**0.049** **(0.261)**	0.034 (0.611)	0 (0.669)
People at risk of poverty or social exclusion in rural zones	0.060 (0.666)	−0.003 (0.622)	0.078 (0.595)	0.184 (0.587)	0 (0.669)

Source: own calculations in MATLAB. Bold numbers are significant.

**Table 12 ijerph-19-11459-t012:** The results of generalized ridge regressions for Malta.

Variables (in Log)	Dependent Variable Unstandardized Coefficients (PP1SD)				
	Self-Perceived Health as Bad or Very Bad (Total)	Energy Import Dependency	Share of Housing Costs in Disposable Household Income	Cooling Days	Heating Days
Arrears on utility bills	**−0.176** **(0.310)**	0 (0.667)	0 (0.667)	0 (0.667)	0 (0.667)
Population unable to keep home adequately warm	0.054 (0.542)	0 (0.667)	0 (0.667)	0 (0.667)	0 (0.667)
People at risk of poverty or social exclusion in cities	−0.174 (0.615)	0 (0.667)	0 (0.667)	0 (0.667)	0 (0.667)
People at risk of poverty or social exclusion in towns	**−0.027** **(0.456)**	0 (0.667)	0 (0.667)	0 (0.667)	0 (0.667)
People at risk of poverty or social exclusion in rural zones	0.148 (0.577)	0 (0.667)	0 (0.667)	0 (0.667)	0 (0.667)

Source: own calculations in MATLAB. Bold numbers are significant.

## Data Availability

The data used were obtained from Eurostat and the World Bank.

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
