# Peer review of "Energy Poverty and Personal Health in the EU"

_ijerph, 2022, doi:10.3390/ijerph191811459_

Round 1

Reviewer 1 Report

Thanks for the opportunity to review the work. This paper focuses on energy poverty and the connection to personal health. The paper assesses the impact of energy poverty on health in the EU-27 countries for the period 2003-2020 using Panel Autoregressive Distributed Lag models and generalized ridge regressions. A separate analysis is made for those EU states with the highest energy import dependency and the implications of the results are discussed.

This paper studies an important research problem. I like the manuscript. I appreciate the significant efforts of the authors on this manuscript. Some comments about the paper are listed as follows. I hope can improve it.

1. The Problems and Motivations of the research about energy poverty and the connection to personal health are not clearly explained. Please improve them to make researchers better understand. The authors also should summarize their contributions clearly in Section 1.

2. In Literature Review Section, the authors need to include and discuss some more recent and relevant works in the area of poverty. Some examples of these works are listed below:
--Differences and Influencing Factors of Relative Poverty of Urban and Rural Residents in China Based on the Survey of 31 Provinces and Cities, https://doi.org/10.3390/ijerph19159015
--Energy poverty, climate shocks, and health deprivations, https://doi.org/10.1016/j.eneco.2021.105338
--The Nexus between Credit Channels and Farm Household Vulnerability to Poverty: Evidence from Rural China, https://doi.org/10.3390/su12073019

3. I didn’t find any content about the details of Equations (1), (2), (3) and (4). Equations (1), (2), (3), and (4) are not well motivated. It is unclear why did you use them. I think it needs more details. Please explain Equations (1), (2), (3) and (4) in detail in the revised manuscript.

4. What is the economic explanation for the results of generalized ridge regressions for Malta in Table 12. Please also use three-line tables for ALL Tables. Now they are informal.

5. What are the limitations of the study and suggested future research directions. Please discuss them clearly in the revised manuscript.

Author Response

First, we would like to thank the reviewer for valuable comments and suggestions to improve the paper.

1. The Problems and Motivations of the research about energy poverty and the connection to personal health are not clearly explained. Please improve them to make researchers better understand. The authors also should summarize their contributions clearly in Section 1.  We have added lines 62-66 and 80-85 in Section 1 to address this.

2. In Literature Review Section, the authors need to include and discuss some more recent and relevant works in the area of poverty. Some examples of these works are listed below:
--Differences and Influencing Factors of Relative Poverty of Urban and Rural Residents in China Based on the Survey of 31 Provinces and Cities, https://doi.org/10.3390/ijerph19159015
--Energy poverty, climate shocks, and health deprivations, https://doi.org/10.1016/j.eneco.2021.105338
--The Nexus between Credit Channels and Farm Household Vulnerability to Poverty: Evidence from Rural China, 

We have reviewed the suggested papers and, respectfully, do not believe that they fit the content of the paper. Therefore, we have not added those suggested papers. We have done an extensive literature search and believe that the most relevant papers are included and cited.

3. I didn’t find any content about the details of Equations (1), (2), (3) and (4). Equations (1), (2), (3), and (4) are not well motivated. It is unclear why did you use them. I think it needs more details. Please explain Equations (1), (2), (3) and (4) in detail in the revised manuscript.

Lines 220-224; 231-233; 241-245; 298-299; 332-339 have all been added to address this comment.

4. What is the economic explanation for the results of generalized ridge regressions for Malta in Table 12. Please also use three-line tables for ALL Tables. Now they are informal.

Lines 534-540 have been added to address this comment.

Additionally, all tables have been redone to try to make them better.

5. What are the limitations of the study and suggested future research directions. Please discuss them clearly in the revised manuscript.

Lines 609-613 have been added.

Reviewer 2 Report

The article illustrates a hot topic in the EU, which has recently assumed more and more relevance considering rise in energy prices due to the Russian-Ukrainian conflict. In particular, the study extensively investigates the relation between energy poverty and health across EU-27 member states.

The background in concise and clear. The overall paper is well written and the structure helps the reader in following logic steps of the methodology.  I would recommend accepting the paper with just few very small revisions:

-        The originality of the study with respect to previous literature could be expressed more clearly.

-        Lines 87-88 seems a repetition of lines 38-40.

-        Several one-page tables interrupt the flow of the reading. Perhaps these could be synthesized in shorter tables, and the full data might be included as Annexes.

Author Response

First, we would like to thank the reviewer for valuable insights, suggestions, and corrections.

The originality of the study with respect to previous literature could be expressed more clearly.

Lines 81-89 have been added to try to address this and improve flow

Lines 87-88 seems a repetition of lines 38-40.

This has been corrected in line 102

Several one-page tables interrupt the flow of the reading. Perhaps these could be synthesized in shorter tables, and the full data might be included as Annexes.

All tables have been redesigned so they are better. It is not possible to make the tables smaller as the type of results don't allow for this. 

Reviewer 3 Report

The article deals with a very important topic and on a subject that is still little explored, even if it is fundamental in the contemporary scenario. The methodological development is well structured and well articulated. The clear and consistent conclusions

Author Response

We thank the reviewer for the time and effort put into our paper. We appreciate the kind words. 

Round 2

Reviewer 1 Report

Some comments about the paper are listed as follows.

1. The article title (New insights) is unreasonable (unfocused, too big). Please improve it.
I do NOT find what are your exactly New insights, and  you only use the words (new insights) in the article title.

2. Sections 6 and 7 (Discussion and Conclusions) sound trivial, the authors should develop them further.

3. All tables and equations are ugly. Please see the tables and equations of published paper in International Journal of Environmental Research and Public Health, and improve them.  Please use three-line table.

4. The related works about energy poverty are insufficient, incomplete and rough. The authors needs to discuss the differences between your work and relevant works. The authors needs to discuss some more recent and relevant works in the area. Some examples of these works are listed below:
--Implications and measurement of energy poverty across the European Union. Sustainability, 8(5), 483.
--Streimikiene, D., Lekavičius, V., Baležentis, T., Kyriakopoulos, G. L., & Abrhám, J. (2020). Climate change mitigation policies targeting households and addressing energy poverty in European Union. Energies, 13(13), 3389.
--Xiao, Y., Wu, H., Wang, G., & Wang, S. (2021). The Relationship between Energy Poverty and Individual Development: Exploring the Serial Mediating Effects of Learning Behavior and Health Condition. International Journal of Environmental Research and Public Health, 18(16), 8888.
--Bukari, C., Broermann, S., & Okai, D. (2021). Energy poverty and health expenditure: evidence from Ghana. Energy Economics, 103, 105565.
--The non-linear effects of energy efficiency gains on the incidence of energy poverty. Sustainability, 13(19), 11055.

5. The format of References is go-as-you-please, please see the published paper of International Journal of Environmental Research and Public Health. Please improve them.

You cannot take the comments lightly and publish this paper in this form with the issues for audience to read.

Author Response

We thank the reviewer for their comments. We have our responses in blue below the reviewer’s comments.

  1. The article title (New insights) is unreasonable (unfocused, too big). Please improve it.
    I do NOT find what are your exactly New insights, and  you only use the words (new insights) in the article title.

The title has been changed to “Energy Poverty and Personal Health in the EU”

2. Sections 6 and 7 (Discussion and Conclusions) sound trivial, the authors should develop them further.

Sections 6 and 7 have been significantly altered. Section 6 has had some material added to reinforce the comments already made and Section 7 has been considerably reworked. This can be seen with the track-changes.

3. All tables and equations are ugly. Please see the tables and equations of published paper in International Journal of Environmental Research and Public Health, and improve them.  Please use three-line table.

The tables and equations have been redone to the style outlined on the website of IJERPH. We revised them previously but did not understand or realize that the reviewer was referring to a table style in Word, so that was a confusion. We did not take the reviewer’s comments lightly, we did not understand that this was a particular style for a table in Word.

4. The related works about energy poverty are insufficient, incomplete and rough. The authors needs to discuss the differences between your work and relevant works. The authors needs to discuss some more recent and relevant works in the area. Some examples of these works are listed below:
--Implications and measurement of energy poverty across the European Union. Sustainability, 8(5), 483.
--Streimikiene, D., Lekavičius, V., Baležentis, T., Kyriakopoulos, G. L., & Abrhám, J. (2020). Climate change mitigation policies targeting households and addressing energy poverty in European Union. Energies, 13(13), 3389.
--Xiao, Y., Wu, H., Wang, G., & Wang, S. (2021). The Relationship between Energy Poverty and Individual Development: Exploring the Serial Mediating Effects of Learning Behavior and Health Condition. International Journal of Environmental Research and Public Health, 18(16), 8888.
--Bukari, C., Broermann, S., & Okai, D. (2021). Energy poverty and health expenditure: evidence from Ghana. Energy Economics, 103, 105565.
--The non-linear effects of energy efficiency gains on the incidence of energy poverty. Sustainability, 13(19), 11055.

The references above have all been added to the paper and the Nawaz reference from the first review was added. Additionally, the contribution of this paper to the literature has been added in several sections of the paper, specifically in the Introduction and Conclusion, but other areas as well. For example, lines 84-100, lines 188-192, and Section 7.

5. The format of References is go-as-you-please, please see the published paper of International Journal of Environmental Research and Public Health. Please improve them.

The references have been formatted to the IJERPH requirements on the website. We believe that something happened with the original submission that the links to Endnote got broken and that caused formatting issues, which you have outlined. This software glitch also caused issues with adding the Nawaz reference from the initial review. We have fixed those (Endnote technical support took 1.5 hours to fix this as they did not know what happened either) and to be sure there won’t be any issues this time we will also submit a pdf to show that the references are in the appropriate style.

You cannot take the comments lightly and publish this paper in this form with the issues for audience to read.
